# Unlocking the Power of GANs in Non-Autoregressive Text Generation

## Abstract

Generative Adversarial Networks (GANs) have been studied in text generation to tackle the exposure bias problem. Despite their remarkable development, they adopt autoregressive structures so suffering from high latency in both training and inference stages. Although GANs have potential to support efficient generation by adopting non-autoregressive (NAR) structures, their explorations in NAR models are extremely limited. In this work, we conduct pioneering study of building language GANs based on NAR structures. We identify two issues that constrain the performance of GAN-based NAR models. Firstly, existing methods of incorporating latent variables provide highly similar representations which cannot describe the diversity of different words in sentences. We tackle this problem by proposing Position-Aware Self-Modulation, providing more diverse and effective representations. Secondly, the attention mechanism in Transformer cannot accurately build word dependencies in the unstable training of GANs, and we adopt Dependency Feed Forward Network to enhance the model capacity in dependency modeling. Armed with these two facilities, we propose a GAN-based NAR model, Adversarial Non-autoregressive Transformer (ANT). The experimental results demonstrate that ANT can achieve comparable performance with mainstream models in a single forward pass and has great potential in various applications like latent interpolation and semi-supervised learning.

## 1 Introduction

Generative Adversarial Networks (GANs) (Goodfellow et al., 2014) denote a powerful family of generative models. Unlike diffusion models (Ho et al., 2020; Saharia et al., 2022) and autoregressive (AR) models (Zhang et al., 2021; Yu et al., 2022) requiring multiple inference steps, GANs can produce high quality samples in a single forward pass, and thus have much lower latency (Sauer et al., 2023; Kang et al., 2023). GANs were first applied to text generation to tackle the notorious exposure bias problem (Bengio et al., 2015), which arises from the discrepancy between training and inference processes in Maximum Likelihood Estimation (MLE)-based AR models. These language GANs (Yu et al., 2017; de Masson d'Autume et al., 2019; Ren & Li, 2023) keep the AR structures and tackle the exposure bias problem by using previously generated words as input in both training and inference stage. When providing consistent training and test processes, they do not support parallel computation and have high latency. The high efficiency nature of image GANs are completely lost in existing language GANs. In theory, the global optimality of GANs is achieved if and only if the learned distributions are exactly same with the real distributions (Goodfellow et al., 2014). More importantly, their convergence does not rely on specific structures. GANs can theoretically obtain high quality samples in a single forward pass. Even so, their explorations in non-autoregressive (NAR) text generation are extremely limited.

In this paper, we unlock the power of GANs in building NAR text generative models. Instead of using AR structures like existing language GANs (Yu et al., 2017; Lin et al., 2017; Che et al., 2017; de Masson d'Autume et al., 2019; Ren & Li, 2023), our model adopts NAR structures supporting high efficiency parallel computation. When benefiting from the generation efficiency, we observe clear gaps between the performance of existing AR language GANs and our NAR language GANs. The main obstacles come from two problems. Firstly, GANs rely on latent variables (which are always from a pre-defined distribution) to support sampling, while existing methods of incorporating latent variables provide highly similar representations. These representations cannot describe

the diversity between words and thus leading to inaccurate sentence generation. Secondly, Transformer (Vaswani et al., 2017), widely used in NAR models (Gu et al., 2018; Ghazvininejad et al., 2019), establishes word dependencies solely through the attention mechanism. However, the dynamic weight assignment process becomes unstable during the fragile training of GANs, causing the loss of word dependencies and ultimately resulting in ungrammatical outputs.

Regarding the first problem, we propose **Position-Aware Self-Modulation** which can provide diverse hidden representations for the model to obtain various words in sentences. For the second problem, we replace the original Feed-forward Network (FFN) module in Transformer to be our proposed **Dependency Feed-forward Network (Dependency FFN)**. Different with the attention mechanism whose dependency is easily lost with poor weight assignment, Dependency FFN provides more stable methods for dependency modeling. Armed with these two facilities, we propose an **Adversarial Non-autoregressive Transformer (ANT)**. The contributions of this work are summarized as follows:

- We conduct pioneering work of building language GANs in NAR structures. To support the generation of various words in sentences, we propose Position-Aware Self-Modulation which can obtain diverse representations to describe the diversity of different words and improve generation quality. Furthermore, Dependency FFN is proposed to support more stable dependency modeling. Different with the attention mechanism, which is easily influenced by the unstable training process of GANs, Dependency FFN can help the model build more accurate dependencies and thus obtain more grammatical results.

- Utilizing these two facilities, we propose a GAN-based NAR text generative model—ANT. Existing language GANs employ AR structures, which leads to high latency due to their reliance on previously generated words. ANT, however, generates all words in parallel and support high efficiency generation. It inherits the advantages of GANs, enabling the generation of high-quality samples in a single forward pass.

- The experimental results demonstrate that ANT achieves performance comparable to existing models, but with significantly lower latency, in both unconditional and conditional generation tasks. Besides, we also explore the potential of ANT in applications like latent interpolation and semi-supervised learning. To the best of our knowledge, it is the first work demonstrating the effectiveness of GANs in building NAR text generative models.

## 2 BACKGROUND

GANs (Goodfellow et al., 2014) are initially transferred to text generation to tackle the notorious exposure bias problem (Bengio et al., 2015). More specifically, existing text generative models adopt autoregressive structures as backbones and use Maximum Likelihood Estimation (MLE) as training objectives. These methods use ground truth as input during training, but reads previously generated words during inference. When the model makes mistakes in generation, these mistakes will be fed into the model as input and the model will be in the state space it has never met during training (Bengio et al., 2015). These mistakes will thus be quickly amplified, leading to a sharp decrease in the quality of the generated samples.

The training of GANs does not need ground truth as input, so they can use generated tokens in both training and inference stage. It tackles the exposure bias problem by providing a consistent generation manner in training and test procedures. In text generation, the generator often models the output word probabilities and sample specific words from these probabilities. This sampling operation, however, is non-differentiable and stops the gradients from being passed through to the generator.

Early study tackles this problem by using either REINFORCE (Yu et al., 2017; Lin et al., 2017; Che et al., 2017; Guo et al., 2018; de Masson d'Autume et al., 2019) or continuous relaxations (Nie et al., 2019). However, REINFORCE Williams (1992) is in high variance, while continuous relaxations like Gumbel-softmax Jang et al. (2017) are biased estimators. Models based on these two methods rely on pre-training techniques to obtain acceptable performance (Ren & Li, 2023).

Another method is to transform words into representations, and train the generator to obtain these representations. This method avoid the non-differentiable sampling operation during training, so the gradients can be passed to the generator directly. Different with REINFORCE and continuous

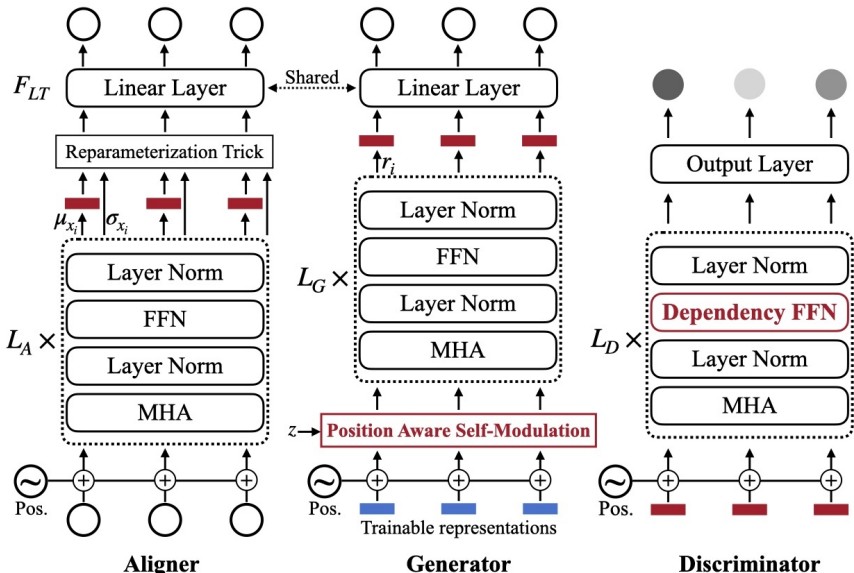

Figure 1: Structure of Adversarial Non-autoregressive Transformer (ANT)

relaxations, which directly model word probabilities, these methods model word representations so they are denoted as representation modeling methods (Ren & Li, 2023). This method can obtain satisfied performance without using any pre-training techniques. We thus build our model based on this method.

NAGAN (Huang et al., 2021), which also adopts GANs to build NAR models, is the most similar work to ours, yet the performance of their model is significantly limited by the biased *straight-through estimator* (Bengio et al., 2013), whereas our model is free from this problem. Besides, our proposed facilities: Position-Aware Self-Modulation and Dependency Feed Forward Network can further boost model performance and they are not explored in previous work.

Furthermore, existing MLE-based NAR models (Gu et al., 2018; Ghazvininejad et al., 2019) suffer from the multi-modality problem Gu et al. (2018), which tends to mix words in different candidates and obtain ungrammatical results. Huang et al. (2022a) reveal that the KL divergence between their learned distributions and the real distributions remains non-negative lower bounds. In theory, the learned distribution cannot be exactly same with the real distributions unless words in sentences are independent to each other (which does not match the real situation). Thus, existing NAR models are mainly developed in several specific tasks, while their explorations in more general tasks (like unconditional generation) are extremely limited. This further underscores the pressing necessity of investigating more promising approaches for building NAR models, thereby enabling their application across diverse domains.

## 3 MODEL

### 3.1 MODEL STRUCTURE

In this paper, we propose an Adversarial Non-autoregressive Transformer (ANT) which generates text in a fully NAR manner. ANT is based on the representation modeling framework (Ren & Li, 2023). As shown in Figure 1, there are three parts in ANT: Aligner, Discriminator and Generator. The aligner maps words into representations, and the generator tries to recover these representations. The discriminator needs to identify whether input representations are from the aligner or the generator. We adopt Transformer (Vaswani et al., 2017) as the backbones of all the three parts to support highly parallel computation. An input is firstly added with a positional encoding and fed into encoder layers. Each encoder layer has a multi-head attention (MHA) module and feed forward network (FFN) module. A layer normalization is added after each module.

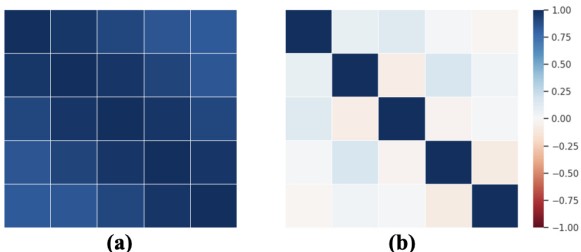

Figure 2: Cosine similarity of the output from (a) Self-Modulation; and (b) Position-Aware Self-Modulation.

The aligner is trained to reconstruct words based on the masked input, which is the same as the training process of BERT (Devlin et al., 2019). Following the previous work which adopts representation modeling methods to train language GANs (Ren & Li, 2023), we use the loss function of variational autoencoder (VAE) (Kingma & Welling, 2014) to train the aligner:

$$L_A = -\mathbb{E}_{z_i' \sim q(z_i'|x_i)}(log p(x_i|z_i')) + KL(q(z_i'|x_i)||p(z_i'))  \tag{1}$$

where $x_i$ is the $i$-th word in the sentence, $z_i'$ is obtained by using reparameterization trick: $z_i' = \mu_{x_i} + \sigma_{x_i} \cdot \mathcal{N}(0, 1)$, and $z_i'$ is transformed back into words with a linear transformation layer $F_{LT}$. Different from cross entropy which maps words into specific points in the representation space, this method describes a region for each word, so representations slightly away from their central points $\mu_{x_i}$ can still be transformed into correct words.

A non-autoregressive generator cannot input previously generated words, so trainable representations are adopted as input. The generator then gives output representations $r_i$ in different positions and uses the same linear transformation layer $F_{LT}$ in the aligner to transform these representations back into words. The discriminator adopts the output representations from the aligner and the generator ($\mu_{x_i}$ and $r_i$) as input. Different from image GANs whose discriminators give a single scaler output for an image, our discriminator gives output for each representation. During training, the aligner will be trained first, and its parameters are fixed during the training of the discriminator and the generator. The representations given by the generator need not be transformed into words in training process, so the gradients from the discriminator can directly pass through to the generator.

Causal masks are adopted in both the discriminator and the generator to break the possible symmetry in the input. We use Wasserstein distance (Arjovsky et al., 2017) as the training objective and adopt Lipschitz penalty (Petzka et al., 2018) to regularize the discriminator. However, there is still a gap between our basic model and existing autoregressive models and we further propose Position-Aware Self-Modulation and Dependency Feed Forward Network (Dependency FFN) to improve model performance.

### 3.2 POSITION-AWARE SELF-MODULATION

An effective sampling method plays a key role in the success of GANs. Transformer based image GANs (Lee et al., 2021) generate different samples by adopting self-modulation (Chen et al., 2019) to incorporate latent variables. Self-modulation assigns the same shift and scale factors to the normalized results in different positions, which leads the representations in various positions to be highly similar even with positional encodings (as shown in Figure 2 (a)). However, the output of the generator (i.e., word representations in different positions) are of high diversities. Similar input representations cannot describe the diversity among different words and thus leading to inaccurate sentence generation.

To tackle this problem, we propose **Position-Aware Self-Modulation**. As shown in Figure 3 (a), this method adopts different mapping layers for the calculations in different positions so as to gain diverse results. In practice, a parallel implementation is adopted to improve the computation effi-

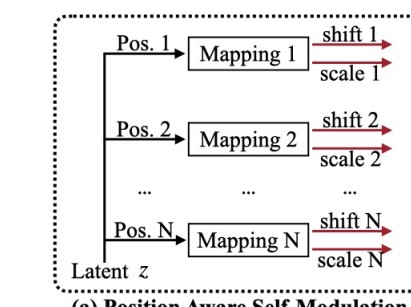 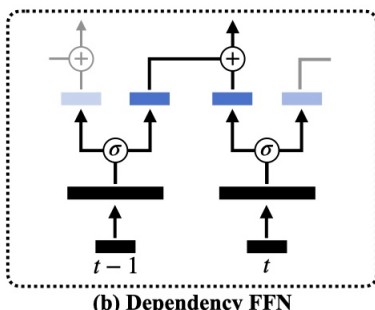

(a) Position Aware Self-Modulation  (b) Dependency FFN

Figure 3: Proposed Facilities in ANT.

ciency, which is:

$$
\begin{pmatrix} \mathbf{h}'_1 \\ \mathbf{h}'_2 \\ \vdots \\ \mathbf{h}'_N \end{pmatrix} = MLP(z)
$$

$$
\mathbf{h}_i = \gamma(\mathbf{h}'_i) \circ LN(\mathbf{x}_i) + \beta(\mathbf{h}'_i)
$$

(2)

where $z$ is the latent variable, $\mathbf{h}'_i$ is the hidden representation in the $i$-th position, $MLP(\cdot)$ is a non-linear transformation whose activation function is GELU (Hendrycks & Gimpel, 2016), $LN(\cdot)$ is the layer normalization, $N$ is the length of the sentence, and $\gamma(\cdot)$ and $\beta(\cdot)$ are linear transformations. In Position-Aware Self-Modulation, representations in different positions are calculated based on unique parameters and have clear differences (as shown in Figure 2 (b)), so as to provide more effective signals to obtain target sentences.

### 3.3 Dependency Feed Forward Network

Transformer (Vaswani et al., 2017) builds word dependencies by dynamically assigning weights in the attention mechanism. This process, however, is unstable under the training of GANs. It will lead the models to lose word dependencies, and finally result in ungrammatical sentences. We tackle this problem by proposing Dependency Feed Forward Network (Dependency FFN) to strengthen the FFN module with the capacity of dependency modeling. The structure of Dependency FFN is shown in Figure 3 (b), and calculated as follows:

$$
\mathbf{s}_t = \sigma(\mathbf{x}_t W_s + b_s)
$$

$$
\mathbf{o}_t = \mathbf{s}_{t-1} W_a + \mathbf{s}_t W_b + b_o
$$

(3)

where $\sigma(\cdot)$ is an activation function which is GELU in this work. With causal masks, $\mathbf{s}_{t-1}$ and $\mathbf{s}_t$ contain the information of first $(t-1)$ and $t$ words, respectively. Using the sum of these two variables can help the model to explicitly build stable dependencies between the $t$-th word and previous $(t-1)$ words in the fragile training process of GANs.

### 3.4 Extension to Conditional Generation

Besides unconditional generation, conditional generation is frequently employed in a variety of tasks. We thus also extend ANT to conditional generation. Given a condition representation $c$, the generator can consider it by shifting the original latent variable $z$. We find that using trainable factors to assign weights to $z$ and $c$ can slightly improve the performance: $\hat{z} = \alpha_1 \circ z + \alpha_2 \circ c$, where $\alpha_1$ and $\alpha_2$ are two trainable variables. For the discriminator, we use the sum of word representations $x_t^d$ and conditional representations $c$ as the input: $\hat{x}_t^d = x_t^d + c$. Then, $\hat{x}_t^d$ is fed into the remaining modules of the discriminator.

Table 1: FED and I. BLEU on the COCO Dataset and EMNLP Dataset (DI: Decoding Iteration).

| Model | DI | COCO Dataset | | EMNLP Dataset | |
|---|---|---|---|---|---|
| | | FED ↓ | I. BLEU ↑ | FED ↓ | I.BLEU ↑ |
| Training Data | - | 0.007 | 35.36 | 0.010 | 20.62 |
| Transformer | O(N) | **0.008** | **34.28** | **0.014** | **19.50** |
| SeqGAN | O(N) | 0.134 | 22.34 | 0.210 | 9.90 |
| RankGAN | O(N) | 0.203 | 22.10 | 0.290 | 10.37 |
| MaliGAN | O(N) | 0.074 | 25.95 | 0.079 | 13.11 |
| LeakGAN | O(N) | 0.132 | 29.43 | 0.125 | 11.59 |
| RelGAN | O(N) | 0.062 | 29.53 | 0.136 | 14.74 |
| ScratchGAN | O(N) | 0.014 | 30.76 | 0.018 | 17.19 |
| InitialGAN | O(N) | 0.013 | 33.06 | 0.025 | 17.74 |
| CMLM | O(k) | 0.016 | 27.65 | 0.062 | **16.67** |
| NAT | O(1) | 0.024 | 26.41 | 0.111 | 11.38 |
| NAGAN | O(1) | 0.084 | 24.98 | 0.748 | 2.01 |
| ANT | O(1) | **0.013** | **31.12** | **0.026** | 15.51 |

## 4 EXPERIMENT

### 4.1 EXPERIMENT SETUP

The experiment covers both unconditional generation and conditional generation to evaluate model performance comprehensively. For the unconditional generation, the task is to generate sentences whose distribution can be as close as to the target sets. We follow the settings of previous work (de Masson d'Autume et al., 2019; Ren & Li, 2023) and use sentences from two datasets: the COCO Image Caption Dataset (Lin et al., 2014)[1] and the EMNLP 2017 News Dataset[2]. The size of training sets of the COCO dataset and the EMNLP dataset are set to be 50,000 and 200,000, respectively. The COCO dataset can support evaluations in short sentence generation, while the EMNLP dataset focuses on long sentence generation. For the conditional generation, we randomly select 100,000 sentences from the Yelp Dataset[3] as training data and use emotion labels (positive or negative) as conditions.

### 4.2 EVALUATION METRICS

The evaluation is conducted at both embedding level and token level. In embedding level, we use Universal Sentence Encoder[4] (Cer et al., 2018) to transform sentences into embeddings. Then, we calculate both **Fréchet Embedding Distance (FED)** (de Masson d'Autume et al., 2019) and **Least Coverage Rate (LCR)** (Ren & Li, 2023) to evaluate the overall similarity and the fine-grained similarity of two distributions, respectively.

In token level, we use **Inverse-BLEU (I. BLEU)** to evaluate model performance in terms of quality and diversity together. Besides, we also draw a curve of **BLEU** (Papineni et al., 2002) and **Self-BLEU** (Zhu et al., 2018) by tuning the temperature of the model (Caccia et al., 2020). In the case of conditional generation, **Accuracy (Acc.)** is also employed to assess whether the models produce sentences that align with the input labels.

### 4.3 COMPARED MODEL

An important compared model is Transformer, which adopts AR structures and is trained on MLE. It is the mainstream model in various text generation tasks. Besides, a number of AR language GANs are also compared: SeqGAN (Yu et al., 2017), RankGAN (Lin et al., 2017), MaliGAN (Che et al.,

---

[1]https://cocodataset.org
[2]http://www.statmt.org/wmt17/
[3]https://www.yelp.com/dataset
[4]https://tfhub.dev/google/universal-sentence-encoder/4

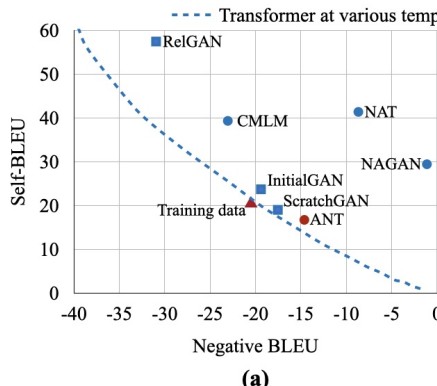 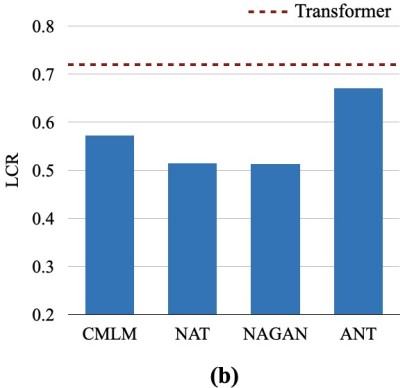

Figure 4: Additional Experimental Results. (a) Model Performance at Various Temperature. (b) Least Coverage Rate.

2017), LeakGAN (Guo et al., 2018), and ScratchGAN (de Masson d'Autume et al., 2019), which are based on *REINFORCE*; RelGAN (Nie et al., 2019), which uses *Gumbel-softmax* to obtain gradients; InitialGAN (Ren & Li, 2023), which does not use the above two method and adopts representation modeling. All the models mentioned-above are AR models whose Decoding Iteration (DI) is $O(N)$ ($N$ is the sequence length).

For NAR models, we compare with another GAN-based model: NAGAN (Huang et al., 2021). Furthermore, two classical MLE-based NAR models are compared in our experiments: Non-autoregressive Transformer (NAT) (Gu et al., 2018) and conditional masked language model (CMLM) (Ghazvininejad et al., 2019). More illustrations about experiment details can be found in Appendix A. We will release our code to the public in the future.

## 4.4 EXPERIMENTAL RESULT

### 4.4.1 UNCONDITIONAL GENERATION

The experimental results of the unconditional generation are shown in Table 1. For the COCO dataset, Transformer gets the best performance in AR models, while ANT is the best one in NAR models. More specifically, ANT obtains 0.013 in FED. This result outperforms a number of AR language GANs and is close to the InitalGAN, which is the best language GANs on the COCO dataset. Similar results can be found in Inverse BLEU (I. BELU). ANT gets 31.12 in I. BLEU and it is much better than other NAR models. The performance of MLE-based NAR models (NAT and CMLM) is far behind the AR models. Existing MLE-based NAR models make use of the characteristic of specific tasks (like strong corresponding relation between input and output in machine translation) to relieve the multi-modality problem Gu et al. (2018). Once they are transferred to more fundamental tasks (e.g., unconditional generation), the inhere problem will be more severe and lead to the obvious decrease of model performance. NAGAN, another GAN-based NAR model, is inferior to all the other models. It shows the limitations of the biased *straight-through estimator*.

For the EMNLP dataset, Transformer is still the best model. ANT outperforms other NAR models in FED, while CMLM can slightly outperform ANT in Inverse BLEU. The iterative decoding mechanism helps CMLM to better process complicated datasets with higher decoding latency. To further discuss their performance in the token level, we follow the suggestions from Caccia et al. (2020), and draw the curve of Self-BLEU and Negative BLEU by tuning the temperature in Transformer. The results are shown in Figure 4 (a). ANT is the only NAR model which can get comparable performance with AR models, while other NAR models (including CMLM) remain behind obviously. Specifically, NAGAN gets extremely low BLEU, which indicates that NAGAN cannot generate fluency sentences. It reveals the difficulties of NAGAN to converge on complicated datasets. Furthermore, we compare Least Coverage Rate (LCR) of Transformer and other NAR models in Figure 4 (b). ANT outperforms CMLM with lower decoding iterations, and it is the only NAR model which can get close performance with Transformer.

Table 2: FED, I. BLEU and Acc. on the Yelp dataset

| Model | DI | FED | I. BLEU | Acc. |
|---|---|---|---|---|
| Training Data | - | 0.008 | 24.18 | 92.47% |
| Transformer | O(N) | 0.011 | 23.04 | 91.73% |
| CMLM | O(k) | **0.015** | 18.35 | 87.85% |
| NAT | O(1) | 0.032 | 11.81 | 83.54% |
| ANT | O(1) | 0.018 | **19.08** | **88.35%** |

### 4.4.2 CONDITIONAL GENERATION

The experimental results of conditional generation are shown in Table 2. Among NAR models, ANT gets comparable performance with CMLM in FED, and achieves higher Inverse BLEU and Accuracy with lower decoding latency. NAT, which also generates samples in one decoding step, is inferior to other models. For the accuracy, ANT gets 88.35% which is the highest one among all the NAR models. ANT can generate sentences consistent with the given labels.

Both the experimental results in unconditional generation and conditional generation demonstrate the effectiveness of ANT. It outperforms a number of AR language GANs. Even comparing with the existing best language GANs, it can still obtain comparable performance in much lower latency. It is thus not necessary to build language GANs in AR structures. Besides, ANT is free from the theoretical limitations in MLE-based NAR models, so it can obtain better performance and denotes a more promising methods in building NAR models.

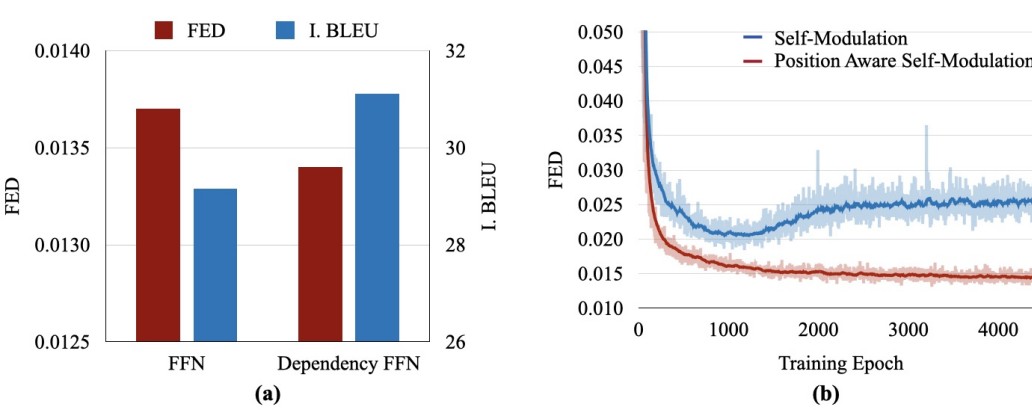

Figure 5: Ablation study of (a) Dependency FFN, and (b) Position-Aware Self-Modulation.

### 4.4.3 ABLATION STUDY

We design two features for ANT: 1) Dependency FFN; and 2) Position-Aware Self-Modulation. Experiments are conducted to demonstrate the effectiveness of these two modules. For Dependency FFN, we compare the performance between Dependency FFN and the original FFN in Figure 5 (a). ANT with Dependency FFN has lower FED and higher Inverse BLEU. It obtains better performance in both the token-level metrics and the embedding-level metrics. These results show that Dependency FFN can help improve model performance by modeling more accurate word dependencies. For Position-Aware Self-Modulation, we compare the training curves with original Self-Modulation with FED in Figure 5 (b). ANT with Position-Aware Self-Modulation converges faster, and finally achieves better performance. Position-Aware Self-Modulation can enhance model performance by providing more diverse and effective representations.

### 4.5 DISCUSSION

One advantage of ANT is that it only requires one decoding step and has high speedup. We compare the speedup of different models in Figure 6 (a). ANT is 14.75 times faster than Transformer. Even

Table 3: Effectiveness of ANT in Semi-supervised Learning (Num.: number of labeled data).

| Method | Num. | P | R | F1 |
|--------|------|-------|-------|--------|
| SL | 500 | 91.28% | 89.06% | 90.15% |
| SSL | | 90.77% | 92.15% | **91.46%** |
| SL | 1000 | 92.42% | 91.33% | 91.87% |
| SSL | | 94.87% | 92.39% | **93.62%** |

comparing with CMLM, it also has much lower decoding latency while obtaining comparable or even better performance. Besides, ANT has great potential in various applications. We explore the potential of ANT in different applications including semi-supervised learning and latent interpolation in the following.

Generative models can be incorporated into semi-supervised learning (SSL) to assist the training of other models. It requires the models to be in high efficiency, since it needs to obtain new data during the training process. We investigate the application of ANT in SSL by incorporating it into the training of a classification model. The classification model is trained to identify emotion labels of sentences in the Yelp dataset. We prepare two training sets. One is composed of 500 labeled data and the other one consists of 1,000 labeled data. The results are shown in Table 3. For the classification model trained on 500 labeled data, its F1 score increases from 90.15% to 91.46%. It gains +1.31% improvement after using SSL. For the model trained 1,000 labeled data, its F1 scores increases from 91.87% to 93.62% in which +1.75% improvement is led by the SSL. The classification models trained in SSL consistently outperform the ones trained in supervised learning (SL). ANT can obtain data following same distributions as the original data so as to help the classification model improve performance.

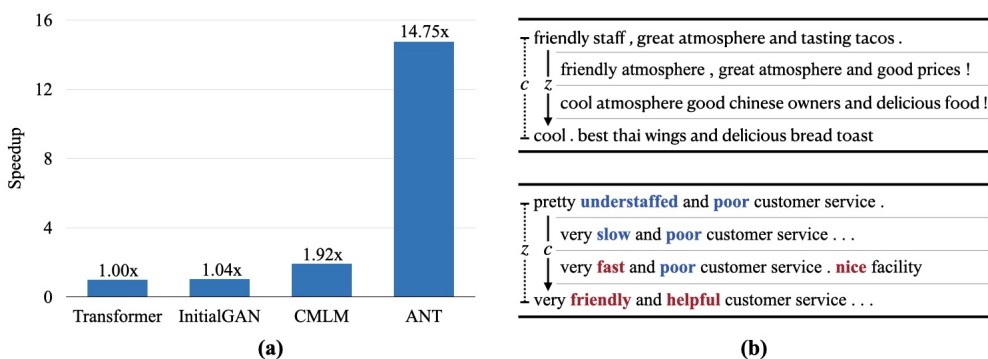

Figure 6: (a) Speedup of Different Models. (b) Case Study of Latent Interpolation.

Besides, ANT enables latent interpolation just like image GANs. There are two latent variables in ANT: $z$, which is sampled from a pre-defined distribution; and $c$, which is a condition representation. We fix one of them and gradually change the other one. The first example in Figure 6 (b) shows the samples given by tuning $z$ with fixed $c$, in which ANT transforms one sentence into another one, with the middle sentences kept understandable. The second example shows the samples given by changing $c$ from the negative representation to the positive representation. ANT gradually transforms negative words into positive ones while keeping the main structure of the sentence. Such latent interpolation is seldomly explored by NAR models, and it may inspire further ideas for related tasks.

## 5 CONCLUSION

In this work, we firstly reveal the limitations in existing language GANs based on AR structures. The global optimality of GANs do not rely on specific structures and NAR structures can obtain similar performance with much lower latency. Then, we reveal two problems in building GAN-based NAR models: 1) Existing methods of integrating latent variables obtain similar representations which

cannot describe the diversity of different words. 2) The attention mechanism in Transformer cannot build word dependencies stably. We tackle these two problems by proposing Position-Aware Self-Modulation and Dependency Feed Forward Network, respectively. Armed with these two facilities, we propose an Adversarial Non-autoregressive Transformer (ANT), a GAN-based NAR model. The experimental results demonstrate that ANT obtains comparable performance as other AR models but with much lower decoding latency. Besides, we also demonstrate the great potential of ANT in various tasks like smooth latent interpolation and semi-supervised learning.

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

## A  EXPERIMENT DETAILS

### A.1  IMPLEMENTATION DETAILS

The layer numbers of the aligner, generator and discriminator are all set to be 4. Their input dimension is 256, and the hidden dimension of FFN / Dependency FFN is 1,024. The head number is set to be 8. We use AdamW (Loshchilov & Hutter, 2019) as the optimizer of the aligner and the weight decay is set to be 1e-5; its learning rate is 0.0001. During the adversarial training, AdamW (Loshchilov & Hutter, 2019) is used as the optimizer of the discriminator whose weight decay is set to be 0.0001; its learning rate is 0.0002 for the COCO and Yelp dataset, and 0.00015 for the EMNLP dataset. We choose Adam (Kingma & Ba, 2015) as the optimizer of the generator and its learning rate is 0.0001. The $\beta_1$ and $\beta_2$ in the optimizers of the discriminator and the generator are set to be 0.5 and 0.9, respectively. The maximum training epoch is set to be 4,500. We implement our model based on Tensorflow[5] (Abadi et al., 2015) and the model is trained on NVIDIA GeForce RTX 3090.

---

[5]`https://www.tensorflow.org`

### A.2 IMPLEMENTATION OF NAT AND CMLM

Both NAT (Gu et al., 2018) and CMLM (Ghazvininejad et al., 2019) are designed for machine translation, so their original structures contains an encoder to encode input in source language. However, there may be no meaningful input in our experiment (e.g., unconditional generation), so this structure cannot be used in the experiment directly. We thus use the idea of VAE to obtain hidden representations, so they can be transferred to the tasks in our experiments.

More specifically, a Transformer-based encoder is adopted to encode the sentences into hidden representations during training. Then, these representations are fed into the decoder to reconstruct the input sentences. During inference, representations sampled from the standard normal distribution will be fed into the decoder, and the decoder will generate sentences based on the sampled representations. For NAT, the representations are fed into decoder as input. For CMLM, the representations are concatenated with the embeddings of input tokens (masked or unmasked words). The iteration number of CMLM is set to be 10 as in previous work (Ghazvininejad et al., 2019; Huang et al., 2022b).

### A.3 EVALUATION METRICS

**Fréchet Embedding Distance (FED)** (de Masson d'Autume et al., 2019) is same with the Fréchet Inception Distance (FID) (Heusel et al., 2017) except for the encoding model. The encoding model is changed to be fit into text generation. We adopts Universal Sentence Encoder following the settings of the previous work (de Masson d'Autume et al., 2019). It is calculated as follows:

$$FED = ||\mu_1 - \mu_2||_2^2 + Tr(c_1 + c_2 - 2(c_1 c_2)^{1/2}) \tag{4}$$

where $\mu_1$ and $\mu_2$ are the mean, and $c_1$ and $c_2$ are the covariance.

**Least Coverage Rate (LCR)** (Ren & Li, 2023) is proposed to be a compliment when the FED of two models are too close to each other, since LCR is more sensitive to the change of data quality (Ren & Li, 2023). Given two sets of sentence $X_i^a \in \mathbb{X}_a$ and $X_i^b \in \mathbb{X}_b$, LCR is calculated as follows:

$$
\begin{aligned}
S_{ij} &= Sim(\mathbf{Emb}(X_i^a), \mathbf{Emb}(X_j^b)) \\
R_a &= \frac{1}{|\mathbb{X}_a|} \sum_{i=1}^{|\mathbb{X}_a|} \delta(\sum_{j=1}^{|\mathbb{X}_b|} S_{ij} \geq \tau) \\
R_b &= \frac{1}{|\mathbb{X}_b|} \sum_{j=1}^{|\mathbb{X}_b|} \delta(\sum_{i=1}^{|\mathbb{X}_a|} S_{ij} \geq \tau) \\
LCR(\mathbb{X}_a, \mathbb{X}_b) &= min(R_a, R_b)
\end{aligned}
\tag{5}
$$

where $X_i^a$ and $X_i^a$ are the i-th and j-th sentences from sentence sets $\mathbb{X}_a$ and $\mathbb{X}_b$), respectively. $\mathbf{Emb}(\cdot)$ is the model used to transform sentences into embeddings (which is Universal Sentence Encoder in this work), $\tau$ is a hyperparameter, $Sim(\cdot)$ is cosine similarity and $\delta(\cdot)$ is a function which returns 1 if input is higher than 0, and 0 for others.

The idea of LCR is to identify whether a specific mode in one set is covered by the sentences in another set or not. Then, it uses the minimum coverage rates as the output, so LCR can be sensitive to two common problems in text generative models: 1) models tend to generate sentences which are out of the real distributions; and 2) the generated sentences are in high similarities.

All the token level metrics (i.e., **BLEU**, **Self-BLEU**, and **Inverse BLEU**) are calculated up to 5 grams following previous work (de Masson d'Autume et al., 2019; Ren & Li, 2023).

