# OpenReview forum: "Unlocking the Power of GANs in Non-Autoregressive Text Generation"
_ICLR.cc/2025/Conference — ICLR 2025 Conference Withdrawn Submission_

### Official Review · Reviewer_rhky · 2024-10-30

**Soundness:** 2
**Presentation:** 2
**Contribution:** 2
**Rating:** 3
**Confidence:** 2

**Summary:**

This work introduced the Adversarial Non-autoregressive Transformer (ANT), a pioneering study of building language GANs based on non-autoregressive (NAR) structures to address the exposure bias problem and reduce latency in training and inference. ANT tackles two key issues: the lack of diversity in latent variable representations by proposing Position-Aware Self-Modulation, and the inaccurate word dependency modeling in Transformers by adopting a Dependency Feed Forward Network. Experimental results show that ANT achieves performance comparable to mainstream models in a single forward pass, with promising applications in latent interpolation and semi-supervised learning.

**Strengths:**

This work pioneers the development of language GANs based on non-autoregressive (NAR) structures, addressing the high latency issues inherent in autoregressive (AR) models. By generating all words in parallel, the Adversarial Non-autoregressive Transformer (ANT) achieves high-efficiency generation, significantly reducing both training and inference times.

The introduction of Position-Aware Self-Modulation and Dependency Feed-forward Network (Dependency FFN) addresses critical challenges in GAN-based NAR models. Position-Aware Self-Modulation enhances the diversity of hidden representations, leading to the generation of more varied and high-quality words in sentences. Dependency FFN improves the stability and accuracy of dependency modeling, resulting in more grammatically coherent outputs compared to traditional attention mechanisms.

**Weaknesses:**

The baselines selected for comparison in the paper are quite outdated, with the chosen non-autoregressive (NAR) models being from 2018, 2019, and 2021. Given the rapid advancements in the field of natural language processing (NLP), it is crucial to compare the proposed model against the most recent and state-of-the-art NAR models to provide a more accurate assessment of its performance. The absence of comparisons with the latest models raises concerns about the relative effectiveness and competitiveness of the proposed approach.

The experiments conducted in the paper are limited to the COCO and EMNLP datasets, which do not provide a comprehensive evaluation of the model's capabilities. To thoroughly assess the performance and robustness of the proposed NAR model, it is essential to test it on a wider range of datasets, including those for machine translation (e.g., WMT), natural language inference (e.g., SNLI), and text summarization. Evaluating the model on these additional datasets would offer valuable insights into its effectiveness across different NLP tasks, particularly in handling longer texts, which is a critical aspect of many real-world applications. The current dataset selection limits the generalizability and applicability of the findings.

If these aspects were addressed with more comprehensive experiments, it would significantly improve the evaluation and increase the overall score of the paper.

**Questions:**

See weaknesses section

---

### Official Review · Reviewer_1HgL · 2024-11-03

**Soundness:** 2
**Presentation:** 2
**Contribution:** 2
**Rating:** 3
**Confidence:** 5

**Summary:**

This paper introduces the Adversarial Non-autoregressive Transformer (ANT), a GAN-based model for efficient text generation. It proposes two main contributions: Position-Aware Self-Modulation and Dependency Feed Forward Network (Dependency FFN). The study claims that ANT achieves comparable performance to mainstream models with significantly lower latency.

**Strengths:**

- The paper is well-structured and provides clear explanations of the proposed methods and their implications, making it easy to follow the authors' reasoning.

**Weaknesses:**

1. The paper may lack a comprehensive comparison with state-of-the-art non-autoregressive models, which is crucial for establishing the significance of the proposed ANT model:  Glancing Transformer (Qian et al, 2021), Fully-NAT (Gu et al. 2021), DA-Transformer (Huang et al. 2022), SUNDAE (Nikolay Savinov et al. 2022), etc.
2. While the paper claims that Position-Aware Self-Modulation enhances generation diversity, it appears to be a common practice to input identical [MASK] tokens plus positional embeddings in NAR models, which also achieve strong performance during decoding. The paper does not provide direct evidence to show that the similar representation approach hinders the model's generation ability or that Position-Aware Self-Modulation offers a significant improvement over this standard practice. This lack of evidence makes it difficult to assess the true impact of this contribution.
3. The Dependency Feed Forward Network is presented as a solution to the instability of word dependencies during GAN training. However, the provided evidence in Figure 5 shows only a marginal improvement, with the gap in FED not exceeding 0.001. Such a small difference raises questions about the practical significance of this improvement, especially considering the computational overhead it might introduce.

**Questions:**

1. Could the authors provide empirical evidence or further analysis to support the claim that Position-Aware Self-Modulation significantly improves generation diversity compared to standard practices in NAR models?
2. How does the Dependency Feed Forward Network provide a clear advantage over traditional FFNs in the context of GAN training, and what experimental results demonstrate this, beyond the marginal improvement shown in Figure 5?

---

### Official Review · Reviewer_Lecz · 2024-11-04

**Soundness:** 2
**Presentation:** 3
**Contribution:** 3
**Rating:** 5
**Confidence:** 4

**Summary:**

This paper explores the application of Generative Adversarial Networks (GANs) in non-autoregressive (NAR) text generation, addressing the limitations of existing GAN-based text generation models that typically rely on autoregressive structures. The authors identify two main issues with current NAR models: the lack of diversity in latent variable representations and the instability of attention mechanisms during GAN training. To tackle these problems, they introduce two useful techniques: Position-Aware Self-Modulation (PASM) and Dependency Feed Forward Network (DFFN). The experimental results demonstrate that ANT achieves comparable performance to the baselines.

**Strengths:**

- The authors propose Position-Aware Self-Modulation (PASM), which provides more diverse and effective latent variable representations, enhancing the model's ability to capture the diversity of different words in sentences.
- To improve the dependency among the decoding procedure, the authors propose Dependency Feed Forward Network (DFFN),  which can lead to better performance.
- The authors conduct extensive experiments to validate the effectiveness of their proposed model, comparing it against mainstream models and demonstrating its competitive performance.

**Weaknesses:**

- The title of the paper is "Unlocking the Power of GANs xxx." Generally, the strength of GANs lies in the training of the generator and discriminator through a game-theoretic mechanism. However, the main focus of this paper is not on GANs but rather on non-autoregressive text generation. I do not believe the power of GANs lies in non-autoregressive modeling.
- While the authors compare their model, ANT, to several state-of-the-art models, it appears they have selectively chosen only strong non-autoregressive baselines that utilize GANs. Other baseline methods, such as Huang et al. (ICML 2022), should also be discussed to strengthen the claims regarding the model's superiority.
- Additionally, the experiments are conducted primarily on specific tasks and datasets, which are somewhat outdated. It would be valuable to assess how well the model generalizes to other text generation tasks, such as summarization, dialogue generation, and long-form text generation.

**Questions:**

refer to the comments

---

### Official Review · Reviewer_c2z6 · 2024-11-05

**Soundness:** 3
**Presentation:** 3
**Contribution:** 3
**Rating:** 3
**Confidence:** 3

**Summary:**

The paper introduces a novel model called Adversarial Non-autoregressive Transformer (ANT) aimed at enhancing the efficiency and performance of Generative Adversarial Networks (GANs) in text generation. Unlike conventional GANs that rely on autoregressive (AR) structures, ANT leverages a non-autoregressive (NAR) framework, allowing for parallel computation and significantly reducing latency in both training and inference. Key contributions include the introduction of Position-Aware Self-Modulation to enhance representation diversity and Dependency Feed Forward Network (Dependency FFN) to improve dependency modeling. Experimental results show ANT's competitive performance with AR models in terms of quality while achieving lower latency.

**Strengths:**

1. This work is pioneering in applying GANs within a non-autoregressive structure for text generation, presenting novel solutions like Position-Aware Self-Modulation and Dependency FFN to tackle inherent limitations in GAN-based text generation.

**Weaknesses:**

1. The datasets chosen for experimental validation are relatively simple, lacking common tasks like translation and summarization, which weakens the persuasiveness of the results.
2. The issue described in line 57, "the dynamic weight assignment process becomes unstable during the fragile training of GANs," lacks references, in-depth analysis, or detailed description of the phenomenon, making it difficult to thoroughly understand this problem.

**Questions:**

Please check the weakness.

---

### Note · Authors · 2024-11-17

I have read and agree with the venue's withdrawal policy on behalf of myself and my co-authors.